# pGlyco 2.0 enables precision N-glycoproteomics with comprehensive quality control and one-step mass spectrometry for intact glycopeptide identification

Ming-Qi Liu[1,2], Wen-Feng Zeng[3,4], Pan Fang[1], Wei-Qian Cao[1], Chao Liu[3], Guo-Quan Yan[1], Yang Zhang[1], Chao Peng[5], Jian-Qiang Wu[3,4], Xiao-Jin Zhang[3,4], Hui-Jun Tu[3,4], Hao Chi[3], Rui-Xiang Sun[3], Yong Cao[6], Meng-Qiu Dong [6], Bi-Yun Jiang[1], Jiang-Ming Huang[1], Hua-Li Shen[1], Catherine C.L. Wong[5], Si-Min He[3,4] & Peng-Yuan Yang[1,2]

The precise and large-scale identification of intact glycopeptides is a critical step in glycoproteomics. Owing to the complexity of glycosylation, the current overall throughput, data quality and accessibility of intact glycopeptide identification lack behind those in routine proteomic analyses. Here, we propose a workflow for the precise high-throughput identification of intact N-glycopeptides at the proteome scale using stepped-energy fragmentation and a dedicated search engine. pGlyco 2.0 conducts comprehensive quality control including false discovery rate evaluation at all three levels of matches to glycans, peptides and glycopeptides, improving the current level of accuracy of intact glycopeptide identification. The N-glycoproteome of samples metabolically labeled with $^{15}N/^{13}C$ were analyzed quantitatively and utilized to validate the glycopeptide identification, which could be used as a novel benchmark pipeline to compare different search engines. Finally, we report a large-scale glycoproteome dataset consisting of 10,009 distinct site-specific N-glycans on 1988 glycosylation sites from 955 glycoproteins in five mouse tissues.

[1] Institutes of Biomedical Sciences and Department of Chemistry, Fudan University, Shanghai 200032, China. [2] Department of Systems Biology for Medicine, Basic Medical College, Fudan University, Shanghai 20032, China. [3] Key Lab of Intelligent Information Processing of Chinese Academy of Sciences (CAS), Institute of Computing Technology, CAS, Beijing 100190, China. [4] University of Chinese Academy of Sciences, Beijing 100049, China. [5] National Center for Protein Science (Shanghai), Institute of Biochemistry and Cell Biology, Shanghai Institutes for Biological Sciences, CAS, Shanghai 201210, China. [6] National Institute of Biological Sciences (Beijing), Beijing 102206, China. Ming-Qi Liu, Wen-Feng Zeng, Pan Fang, Wei-Qian Cao and Chao Liu contributed equally to this work. Correspondence and requests for materials should be addressed to C.C.L.W. (email: catherine_wong@sibcb.ac.cn) or to S.-M.H. (email: smhe@ict.ac.cn) or to P.-Y.Y. (email: pyyang@fudan.edu.cn)

Protein glycosylation is a heterogeneous post-translational modification (PTM) that generates greater proteomic diversity than other PTMs[1–3]. Certain glycosylation patterns in proteins give rise to functional variance, with far-reaching consequences for health-disease issues, immunological disorders, toxic effects, microbial invasion and other highly important processes[2–4]. Precise and large-scale characterization of protein glycosylation at the site-specific level and the proteome scale is critical for understanding these biological functions. Currently, analysis by liquid chromatography coupled with tandem mass spectrometry (LC-MS/MS) of intact glycopeptides is often the method of choice in site-specific glycoproteomic studies[4–8]. Many explorations and advances in MS-based profiling of intact glycopeptides have been reported[5, 6]. However, precision glycoproteomic studies remain challenging because of the enormous complexity of glycosylation related to factors such as macro- and micro-heterogeneity, the lower abundance and ionization efficiency of glycopeptides relative to regular peptides[4, 5]. Therefore, the overall throughput, the data quality and accessibility of intact glycopeptide identification are considerably lower than those of routine proteomic studies.

To the best of our knowledge, there are three critical limitations of existing methods for intact glycopeptide profiling. First, high-throughput MS acquisition and interpretation pipelines that can perform a confident analysis of both glycans and peptides are lacking. Combining complementary information from multiple sample processing strategies (e.g., analysis of intact glycopeptides, deglycopeptides and released glycans), different MS/MS fragmentations and various software tools (e.g., using different search engines to identify the glycan and peptide) in intact glycopeptide analysis is quite common[9–23], thus compromising the overall throughput and quality[4–6]. Comparatively, in routine proteomic studies, regular peptide identification can be confidently achieved using a single MS/MS spectrum and a search engine. Second, comprehensive quality control has not been developed and integrated into search engines for intact glycopeptide identification, with a false discovery rate (FDR) evaluation needed on all three levels of matches to glycans, peptides and glycopeptides. Third, reliable and general validation methods of spectral interpretations are lacking. Although the current search engines can usually employ some quality control methods[10–17, 24–29], severe underestimation of the FDR has been reported[16].

To address these limitations, we performed extensive analyses and developed novel methods, including a high-throughput MS acquisition method based on optimized MS/MS collision parameters, which generates comprehensive fragments of an intact glycopeptide in a single spectrum. We also propose a dedicated search engine, pGlyco 2.0, that not only fully utilizes the comprehensive fragments in the spectrum but also performs comprehensive quality control over the glycopeptide-spectrum matches (GPSMs). To evaluate the accuracy of our pipeline, a new and quantitative analysis method using metabolically labeled glycoproteome samples was specifically designed to validate glycopeptide identification. Finally, we report a large-scale glycoproteome dataset consisting of 10,009 distinct site-specific N-glycans in five mouse tissues and compare our method with the latest method of comprehensive glycosylation analysis.

## Results

### Development of a MS/MS method and a search engine.
We extensively analyzed the MS/MS fragmentation behavior of glycopeptides to determine the optimum acquisition method[23, 30, 31]. An optimum MS/MS acquisition should generate the most comprehensive fragments in a single spectrum for each intact glycopeptide, including both the glycan and the peptide fragments. A mixture of five standard glycoproteins was analyzed by LC-MS/MS on an Orbitrap Fusion instrument using various MS/MS collision parameters, including collision-induced dissociation (CID) and higher-energy collision dissociation (HCD), each with nine different energies, as well as electron transfer dissociation (ETD) coupled with either CID or HCD (ETciD/EThcD) (Supplementary Notes 1 and 2). Different collision energies in HCD-MS/MS can produce complementary fragments of the glycan and peptide (Supplementary Note 1). CID-, ETD-, ETciD- and EThcD-MS/MS generated fewer fragment ions than HCD-MS/MS (Supplementary Note 1). We then simulated different combinations of HCD-MS/MS under stepped collision energies (SCE). SCE-HCD-MS/MS under 20–30–40% energies generated the most informative and abundant fragment ions for both the glycan and peptide of a glycopeptide in a single spectrum (Supplementary Note 1). An example glycopeptide spectrum obtained under the optimized SCE-MS/MS conditions is illustrated in Fig. 1a, along with a spectrum obtained under the default single-energy HCD-MS/MS conditions for the same glycopeptide (Fig. 1b). The SCE-HCD-MS/MS method used in our study is high throughput and can generate the most comprehensive fragments of an intact glycopeptide reported to date (on an Orbitrap Fusion instrument). We also tested the SCE-HCD-MS/MS method on a Q Exactive instrument. Very similar spectra of the same glycopeptide were produced on these two different instruments (Supplementary Fig. 1), reflecting the general applicability of our MS/MS acquisition method. Importantly, these two MS instruments could only provide a three-step SCE in one spectrum, and SCE-HCD-MS/MS with more flexible collision energy settings could improve glycopeptide analysis (Supplementary Fig. 2).

A dedicated search engine pGlyco 2.0 was then developed to fully utilize the abundant information in SCE-HCD-MS/MS spectra and to confidently perform comprehensive quality control on GPSMs (Fig. 2). pGlyco 2.0 performed an integrated open search of each spectrum: a spectrum was first scored against the glycome database to identify the glycan candidates and then scored against the proteome database to identify the peptide candidates (Methods). Comprehensive quality control, the most important feature of pGlyco 2.0, includes FDR analysis of glycans and peptides, as well as a new model for glycopeptide FDR estimation (Methods). In addition, all identified spectra were automatically annotated and displayed by the software tool gLabel embedded in pGlyco 2.0, which facilitates manual verification and data analysis (e.g., Fig. 1a). Our proposed workflow integrates a fine-tuned MS/MS acquisition method with a search engine dedicated to the comprehensive quality control of intact glycopeptide identification (Fig. 2). At the same time, this workflow requires only a single LC-MS/MS run for glycopeptide spectrum collection, thus allowing high-throughput and highly accurate glycoproteomic analysis.

### Analysis of standard glycoprotein mixture.
We applied the pGlyco 2.0 workflow to the same standard glycoprotein mixture used to optimize the MS/MS fragmentation. We identified nearly two hundred N-glycopeptides in the five glycoproteins with 1% FDR for the GPSMs (Supplementary Data 1). The glycans identified at multiple glycosylation sites in this data set are shown in Fig. 3. Annotated spectra corresponding to identified glycopeptides shown in Fig. 3 can be downloaded in Supplementary Files. Our results were consistent with previously reported glycosylation data for these glycoproteins (Supplementary Note 3). Notably, we used the complete glycoproteome database (the complete databases of both the proteome and glycome for one or more species) to analyze the standard glycoprotein mixture (Methods),

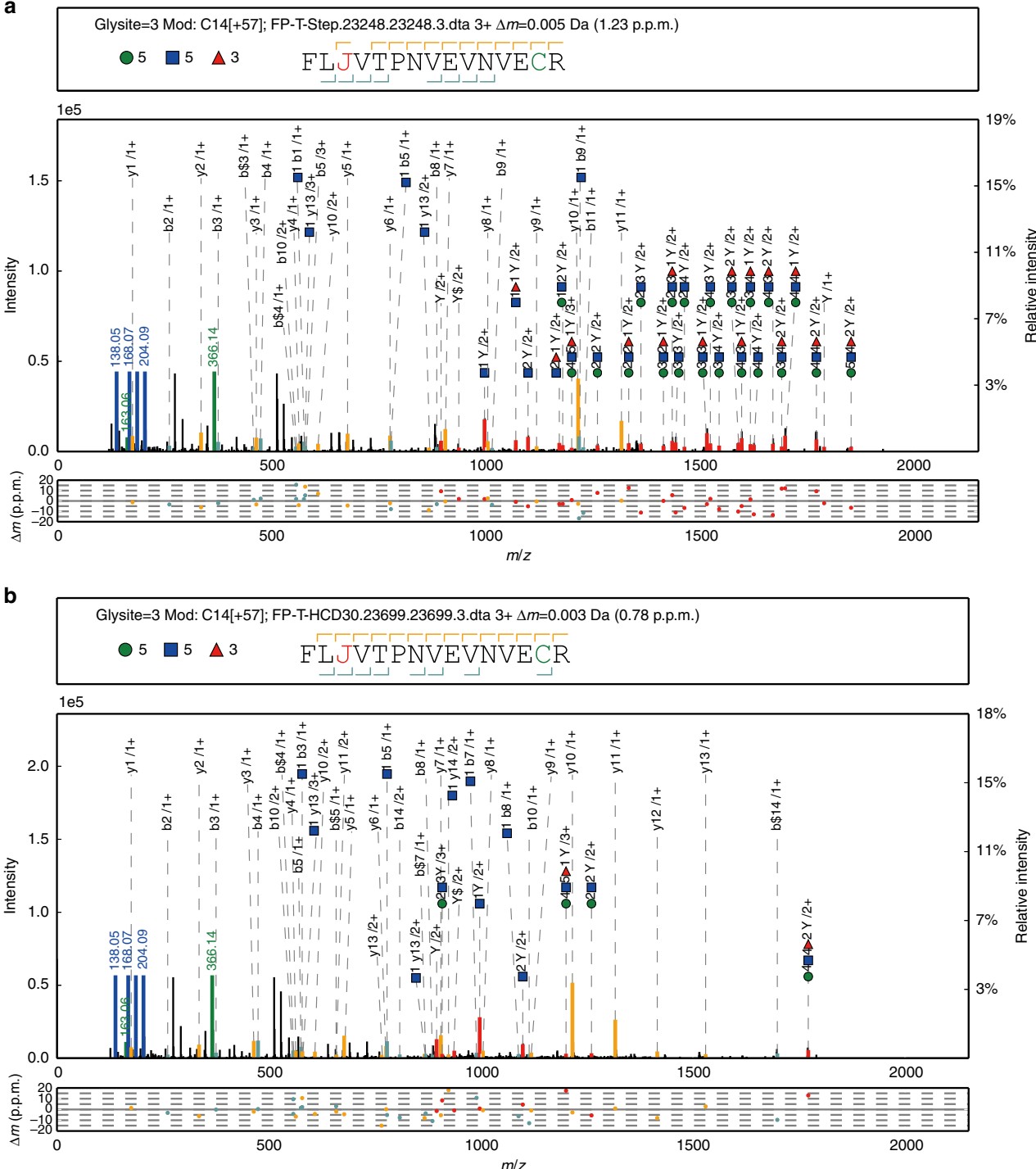

**Fig. 1** Demonstration of the optimized MS/MS collision parameters for an intact glycopeptide. **a** Intact glycopeptide spectrum obtained using the optimized stepped-energy HCD-MS/MS method. **b** Spectrum obtained using the default single-energy HCD-MS/MS method for the same glycopeptide shown in **a**. The design of the upper box above each spectrum consists of the glycosylation site (glysite), modification (mod), spectrum name, precursor mass deviation, glycan composition and peptide sequence with 'J' indicating the N-glycosylation site. The glycan symbols are as follows: green circle for Hex, blue square for HexNAc and red triangle for fucose. Peak annotation is shown in the middle box: *green* and *blue* peaks represent the fragment ions of the glycan moiety or diagnostic glycan ions, and *red* peaks represent the Y ions from glycan fragmentation. For clarity, the scale of the relative intensity is automatically adjusted based on the highest peak between 700 and 2000 Th. Mass deviations of the annotated peaks are shown in the lower box

suggesting that our proposed workflow can be used to analyze glycopeptides in complex samples.

**FDR validation workflow**. To validate the FDR reported by our search engine and other software tools, we established a

quantitative analysis pipeline using metabolically labeled glycoproteome samples. Equal amounts of unlabeled and [15]N- and [13]C-labeled yeast proteins were pooled and then analyzed in one LC-MS/MS run (Fig. 4a). The spectra of intact glycopeptides were collected using the SCE-HCD-MS/MS method

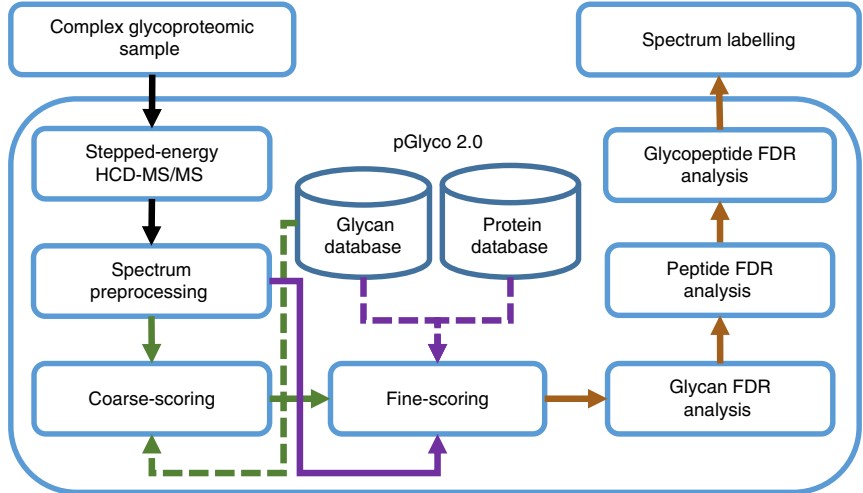

**Fig. 2** Design of a dedicated software for intact glycopeptide interpretation

and then interpreted by the search engines against the same glycoproteome database. A combination of yeast glycome and proteome databases was used as the target glycoproteome database, while a combination of mouse glycome and proteome databases was used as the entrapment database (Fig. 4a). Next, the validity of the reported GPSMs was analyzed using two search engine-independent methods (Fig. 4b): the isotope-based FDR and the entrapment-based FDR. In the isotope-based FDR method, the mass difference between an unlabeled glycopeptide and its $^{15}$N/$^{13}$C-labeled counterpart correlates with the number of their nitrogen or carbon atoms, which is probably different between the true positive and false positive glycopeptides. Therefore, the failure to find an unlabeled and labeled pair in the full mass scans (MS1) leads to a false positive identification (Fig. 4b and Online Methods). In the entrapment-based FDR method, the mouse glycome and proteome databases were used as the entrapment databases. Any GPSM with either a mouse-only glycan or mouse-only peptide was considered a false positive (Fig. 4b and Methods). Note that the aforementioned two methods of FDR estimation are search engine-independent and can therefore be used to compare the performance of different search engines.

**FDR validation results**. We compared the validity of glycopeptide identification reported by different search engines using the proposed workflow for $^{15}$N/$^{13}$C-labeled yeast glycoproteome samples. pGlyco 2.0 successfully estimated the GPSM FDR, while Byonic, a routinely used search engine, severely underestimated the GPSM FDR when searching against the complete glycoproteome database. Byonic was selected for comparison because it was the only search engine that could perform a generic database search of our data against a complete glycoproteome database (we tested many existing search engines with our data, see Supplementary Note 4). We analyzed the GPSMs reported by pGlyco 2.0 and Byonic, and then evaluated the validity using the two aforementioned FDR estimation methods (Fig. 4b and Methods). Examples of the detailed procedure are shown in Fig. 5a, b: for the same spectrum, pGlyco 2.0 and Byonic reported different GPSMs (see the annotated spectra in Supplementary Fig. 3). The glycan of the glycopeptide reported by pGlyco 2.0 belonged to the yeast glycome, and both $^{15}$N- and $^{13}$C-labeled precursors agreed with the glycopeptide composition (Fig. 5a). Alternatively, the glycan of the glycopeptide reported by Byonic belonged to the mouse-only glycome, and neither the $^{15}$N- nor $^{13}$C- labeled precursors agreed with the glycopeptide composition (Fig. 5b), indicating

that the glycopeptide reported by Byonic here is likely a false positive.

The isotope-based FDR and the entrapment-based FDR for the GPSMs identified by pGlyco 2.0 in three LC-MS/MS runs were 0.97% and 0.2%, respectively, which were below the preset criterion of a 1% FDR. The two FDRs for the GPSMs identified by Byonic in the same LC-MS/MS data were 19.4 and 23.9%, respectively (Fig. 5c). Note that both pGlyco 2.0 and Byonic reported GPSMs with a 1% FDR under their own criteria. Clearly, pGlyco 2.0 can successfully estimate the GPSM FDR, while Byonic might severely underestimate this metric. To analyze the major source of false identification reported by Byonic, we adjusted the score threshold in the glycopeptide identifications reported by Byonic (Fig. 5d). As the score threshold increased, the FDR deduced from mouse-only peptides dropped to 0%, while the FDR deduced from mouse-only glycans remained above 20%, indicating that peptide sequence-based scoring and quality control alone could result in a high FDR in the glycan part of glycopeptide identification. Therefore, scoring and quality control on all three levels of matches, i.e., to glycans, peptides and glycopeptides, should be a routine procedure in site-specific glycosylation studies.

**Optimization of LC-MS/MS parameters for large-scale study**. After determining the accuracy of our workflow, we conducted a large-scale and precise N-glycoproteome analysis of five mouse tissues. We first optimized several important LC-MS/MS parameters for intact glycopeptide analysis, including the MS/MS accumulation time, total LC time, sample loading volume and reproducibility between multiple runs (see detailed data in Supplementary Note 5). Interestingly, the optimal MS/MS accumulation time for the identification of glycopeptides (250 ms) and regular peptides (default value of less than 50 ms)[32] by SCE-HCD-MS/MS greatly differed. The preferred total LC time is 6 h (the benefit of a longer LC time is small). For mouse tissues, starting material consisting of 100 μg of protein before enrichment was reasonable for a single LC-MS/MS run. We also analyzed the reproducibility between multiple runs. The reproducibility for a glycopeptide was lower than that for a regular peptide under the same conditions, mainly due to the microheterogeneity of the glycopeptide and the ionization suppression of the regular peptide. Our data suggest that five replicate LC-MS/MS runs may be a preferred approach for large-scale studies of intact glycopeptide.

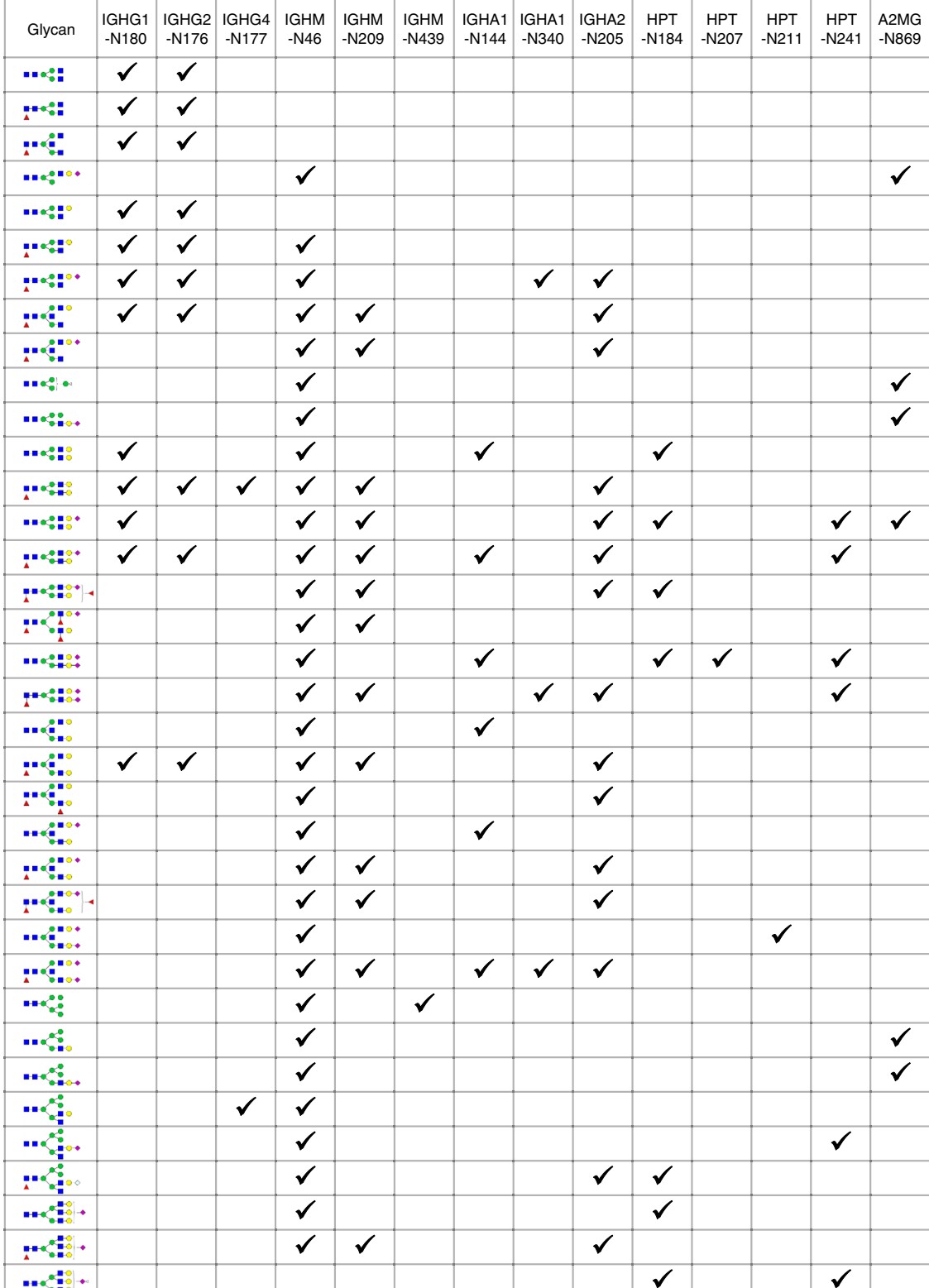

**Fig. 3** Analysis of a standard glycoprotein mixture. Protein names and glycosylation sites are listed in the first row, and glycans are listed in the first column. Identified site-specific glycans are ticked in the table. Glycans identified at more than one glycosylation site are shown here

**Large-scale N-glycopeptide analysis of mouse tissues**. We analyzed the intact N-glycopeptides in five mouse tissues (brain, heart, kidney, liver and lung). In total, 79,930 glycopeptide spectra were identified with a 1% GPSM FDR, corresponding to 10,009 distinct site-specific N-glycans on 1988 glycosylation sites from 955 glycoproteins (Supplementary Data 2–5). The database

search parameters and FDR estimation used in this large-scale analysis were the same as those used in the FDR validation for the yeast sample (Methods). Therefore, we were confident that pGlyco 2.0 achieved a 1% GPSM FDR. The average absolute mass deviation of all identified spectra was 0.84 p.p.m. (Fig. 6a). A C18 LC column with a long gradient showed impressive separation

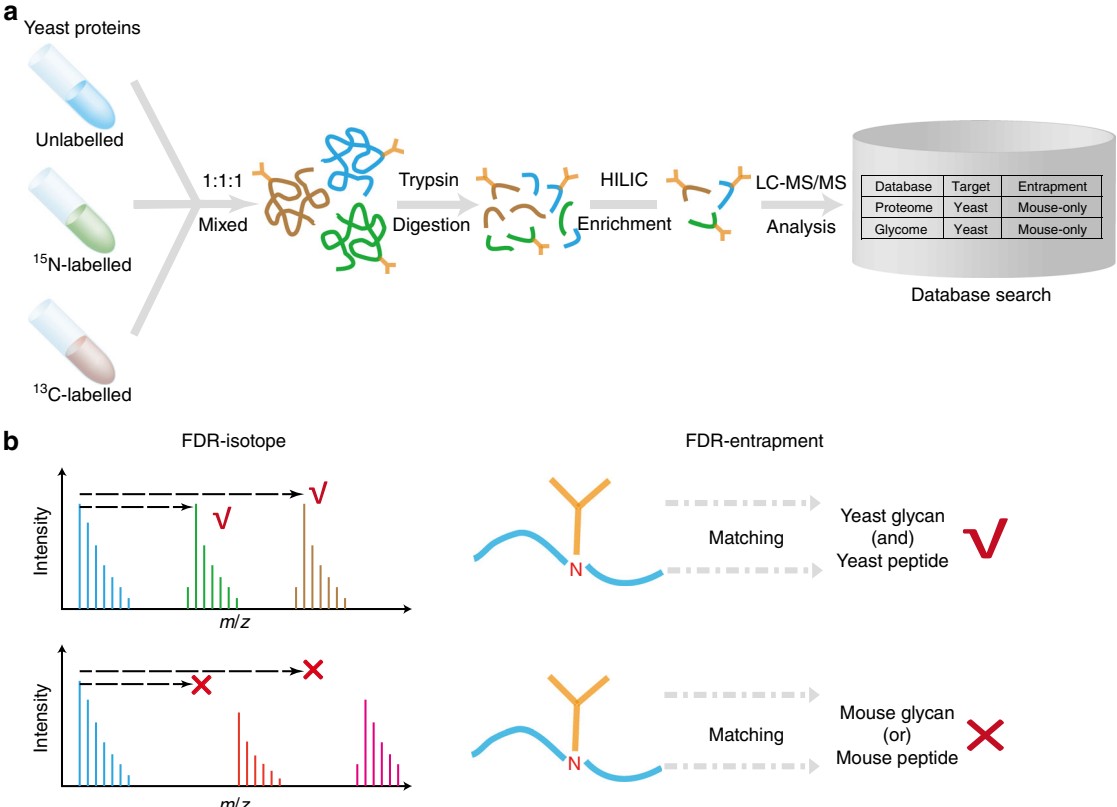

**Fig. 4** FDR validation workflow. **a** Validation workflow. **b** Two search engine-independent FDRs using information from isotope labeling and an entrapment database, respectively

performance for glycopeptides with microheterogeneity (glycopeptides with the same peptide backbone but different glycans). The retention times of glycopeptides with different numbers of sialic acids did not overlap, while other monosaccharides has a lesser but considerable effect on the retention time shift of glycopeptides (Fig. 6b and Supplementary Fig. 4).

We manually compared the SCE-HCD-MS/MS spectra of glycopeptides and regular peptides and found that the glycopeptide spectra had a much larger dynamic range of peak intensities of matched fragments within a spectrum. To verify this finding, we analyzed the effect of different peak filtration thresholds in glycopeptide identification and fine-tuned our search engine accordingly. The filtration threshold for the top 50 peaks met the demand for regular peptide identification. However, the top 300 peaks were required to obtain the optimum threshold for glycopeptide spectra generated using SCE-HCD-MS/MS (Supplementary Fig. 5). In addition, pGlyco 2.0 identified more than 200 chimeric glycopeptide spectra (see spectral annotations in Fig. 6c, the corresponding MS1 profiles in Fig. 6d, and more examples in Supplementary Figs 6 and 7).

**Analysis of the glycosylation profile in mouse tissues.** Different mouse tissues showed distinct glycosylation patterns. Correlation analysis showed that brain tissue demonstrated the most distinctive glycosylation profile compared to the other tissues, while heart and lung tissues were the most similar pair of the five tissues (Fig. 7a and Supplementary Fig. 8). Fucose-containing glycopeptides varied from 7.4% (liver) to over 50% (brain and kidney) (Fig. 7b). There were trace amounts of NeuGc-containing glycopeptides in brain (18.8% for NeuAc and 0.1% for NeuGc), which agrees with a previous finding[33], and an opposite distribution of these two sialic acids was observed in liver (0.2% for

NeuAc and 23.8% for NeuGc) (Fig. 7b). Sialic acid releasing and labeling methods were used to validate the relative abundances of NeuAc and NeuGc in mouse brain and liver (Supplementary Fig. 9). We compared the performances of pGlyco 2.0 and Byonic using the mouse brain data (Supplementary Note 6). pGlyco 2.0 identified more glycopeptides than Byonic, while Byonic reported many glycopeptides with NeuGc in the mouse brain, which contradicts the existing glycomic knowledge[33].

The overall overlap of glycopeptides among tissues was surprisingly small. For example, only 1% of total glycopeptides coexisted in the five tissues (Fig. 7c). The overlapping glycosylation sites of the glycoproteins among tissues were significantly higher than those of the glycopeptides, reflecting the diversity of site-specific glycosylation (Fig. 7c). Moreover, 102 of the 107 (95.3%) the glycopeptides found in all five tissues contained high-mannose glycans. A highly glycosylated protein Q3V3R4 (integrin alpha-1) was selected to demonstrate the tissue specificity of glycosylation (Fig. 8): we identified 131 different site-specific N-glycans on this protein and illustrate the tissue specificity of these N-glycans in the figure. The diversity of glycosylation on the same protein may contribute to its functions in different tissues. Annotated spectra corresponding to identified glycopeptides shown in Fig. 8 can be downloaded in Supplementary Files. In addition, we found that many site-specific glycans were expressed in a tissue-specific manner (Supplementary Table 1). For example, we identified 370 different site-specific N-glycans on protein A2ARV4 (low-density lipoprotein receptor-related protein 2) in mouse kidney, while none of these site-specific N-glycans was found in mouse brain, heart or liver.

**Comparison with existing large-scale glycoproteome research.** We compared our method with one of the most exciting

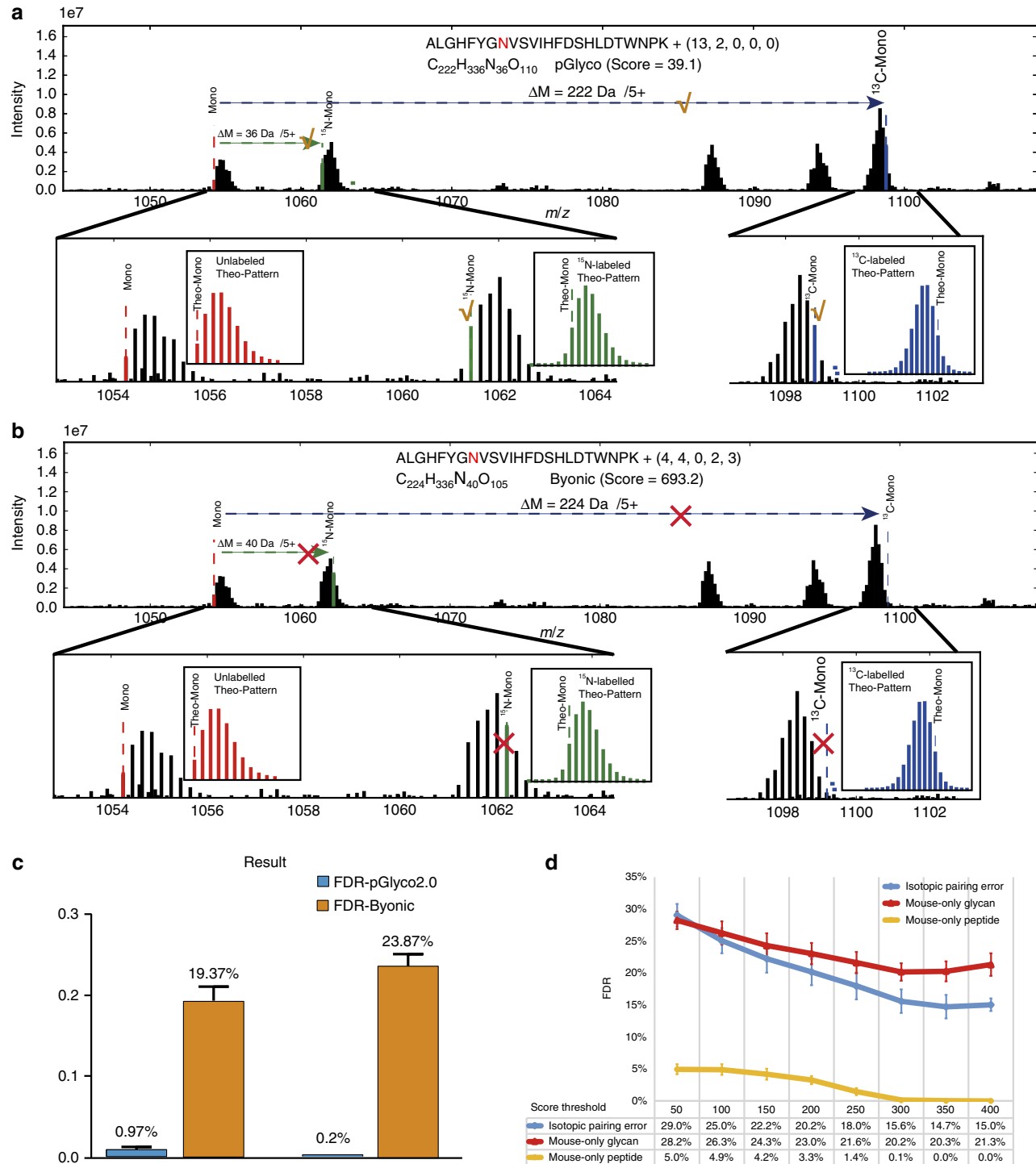

**Fig. 5** FDR validation results. **a**, **b** Isotopic peak pairing analysis of different glycopeptide identifications by pGlyco 2.0 and Byonic for the same spectrum. Both [15]N- and [13]C-labeled precursors agreed with the glycopeptides reported by pGlyco 2.0 **a**. Analysis of the heavily labeled precursor reported by Byonic in MS1. Neither the [15]N- nor [13]C-labeled precursors agreed with the glycopeptides reported by Byonic **b**. **c** FDRs of pGlyco 2.0 and Byonic for three LC-MS/ MS runs. **d** FDR analysis of the glycopeptide identifications reported by Byonic under different score thresholds

advancements in glycoproteomics, NGAG[34]. The major contribution of NGAG is an ingenious enrichment method that interprets deglycopeptides, glycans and intact glycopeptides in a well-controlled manner. NGAG requires two to three separate LC-MS/MS runs for the spectrum collection of deglycopeptides, glycans and glycopeptides in practice, while our workflow can perform a large-scale glycoproteomic study with only a single LC-MS/MS run. The SCE-HCD-MS/MS spectra in our study

contained more comprehensive glycopeptide fragments than the conventional HCD-MS/MS spectra reported in NGAG (Supplementary Fig. 10). Our search engine pGlyco 2.0 showed better performance than the search engine GPQuest used in NGAG: pGlyco 2.0 identified 97% more glycopeptide spectra than GPQuest in the analysis of the same MS/MS data from NGAG[34] under the same peptide FDR (Supplementary Fig. 11 and Methods). Another important advantage of pGlyco 2.0 is that a

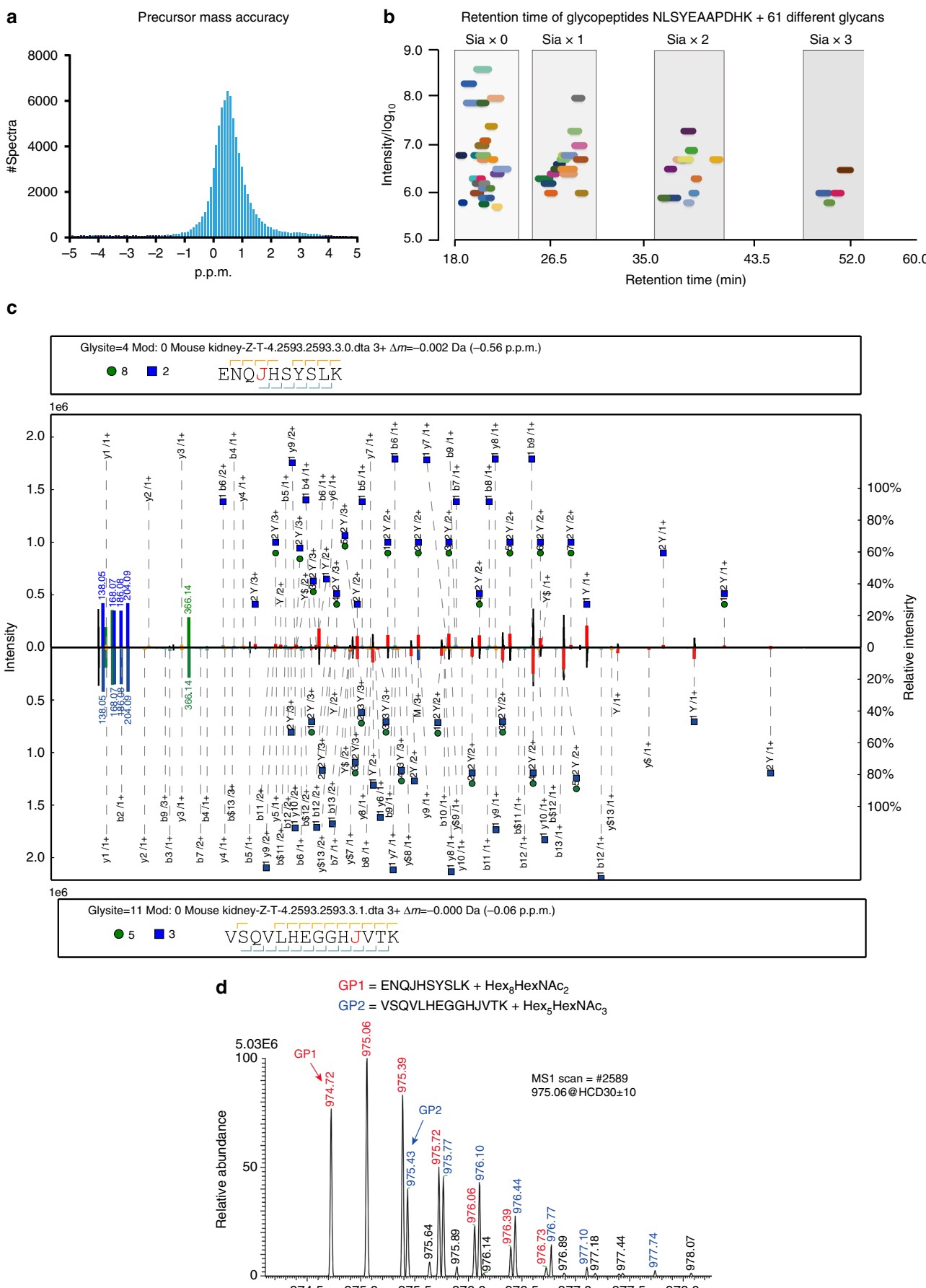

**a** Precursor mass accuracy

**b** Retention time of glycopeptides NLSYEAAPDHK + 61 different glycans

**c**

Glysite=4 Mod: 0 Mouse kidney-Z-T-4.2593.2593.3.0.dta 3+ Δ*m*=−0.002 Da (−0.56 p.p.m.)

● 8  ■ 2   ENQJHSYSLK

Glysite=11 Mod: 0 Mouse kidney-Z-T-4.2593.2593.3.1.dta 3+ Δ*m*=−0.000 Da (−0.06 p.p.m.)

● 5  ■ 3   VSQVLHEGGHJVTK

**d** GP1 = ENQJHSYSLK + Hex$_8$HexNAc$_2$
GP2 = VSQVLHEGGHJVTK + Hex$_5$HexNAc$_3$

MS1 scan = #2589
975.06@HCD30±10

complete proteome database can be used directly in glycopeptide identification, while GPQuest required the sample-dependent construction of a deglycopeptide spectra library or database. We also compared the performance of pGlyco 2.0 for different database sizes: the complete human proteome database and the smaller deglycopeptide database (Supplementary Fig. 11 and Methods). The performance of pGlyco 2.0 with different database sizes is similar. In summary, compared with NGAG, the strengths of our method include higher efficiency and better quality in glycopeptide identification, plus a novel benchmark pipeline for result validation.

In addition, we compared our mouse glycoproteome data with previously reported glycosylation site data[35], which had not undergone analysis for intact glycopeptides. In our data, 85% of the identified glycosylation sites were previously reported (Supplementary Fig. 12). These analyses suggested that pGlyco 2.0 can perform high-quality and high-throughput glycopeptide identification when searching against a complete glycoproteome database.

**Application to O-glycopeptide analysis**. In addition to N-glycosylation analysis, we applied our method to the identification of O-glycosylation. Analysis of a standard glycoprotein, asialofetuin, showed that SCE-HCD-MS/MS provided abundant fragment ions for O-glycopeptides (Fig. 9). Using pGlyco 2.0, we identified many N- and O-glycopeptides in asialofetuin (Supplementary Fig. 13), indicating that our workflow should be suitable for both N- and O-glycosylation analysis. However, we did not observe many mucin-type O-glycopeptides in the mouse tissues, possibly because of the inherent low-abundance O-glycosylation in these samples.

## Discussion

In conclusion, we present a dedicated workflow that combines a fine-tuned, easily adopted MS protocol and a dedicated search engine, pGlyco 2.0, that are ready for use in the precise N-glycoproteomic analysis of complex samples. Comprehensive quality control (FDR evaluation of matches to glycans, peptides and glycopeptides) was integrated into pGlyco 2.0, overcoming the previous limitation related to accuracy in intact glycopeptide identification. By contrast, most search engines for glycopeptide identification only perform quality control on either glycans or peptides. We further validated the accuracy of our glycopeptide identification by a novel analysis of isotopically labeled yeast glycoproteome samples, representing the first use of $^{15}N/^{13}C$ metabolic labeling for the validation of glycopeptide identification. More importantly, our proposed validation method can be used as a general benchmark pipeline for the performance comparison of different search engines, which is lacking in the field of glycopeptide identification. Using our workflow involving SCE-HCD-MS/MS and pGlyco 2.0, large-scale and high-throughput glycoproteomic studies of complex samples can be conducted with high accuracy.

## Methods

**Preparation of standard proteins**. IgG/IgM/IgA from human serum, and hap-toglobin/α-2-macroglobulin from pooled human plasma were purchased from Sigma-Aldrich. Equal amount of the five glycoproteins was pooled as a mixture of standard glycoproteins. Asialofetuin from fetal calf serum used in O-glycopeptide analysis was purchased from Sigma-Aldrich.

**Preparation of yeast samples**. Ammonium chloride ($^{15}N$, 99%) and D-glucose (U-$^{13}C6$, 99%) were purchased from Cambridge Isotoe Laboratories. The fission yeast, *Schizosaccharomyces pombe* (*S. pombe*) strains were kindly provided by Professor Meng-Qiu Dong from National Institute of Biological Sciences (Beijing). The yeast cells were grown in Edinburgh minimal medium at a temperature of 30 °C. Cell growth was followed by measuring the optical density at 600 nm ($OD_{600}$). In the case of the $^{15}N$ labeling experiment, the exact protocol for cell growth was followed, but $^{15}NH_4Cl$ was used as the nitrogen source. Instead, in the case of the $^{13}C$ labeling experiment, normal $^{14}NH_4Cl$ was used but fully labeled $^{13}C$ glucose was used as the carbon source. The cells were incubated with the enriched media for at least 24 h (eight generations) to complete $^{15}N$ and $^{13}C$ labeling. Then the cells were harvested by centrifugation at $1000 \times g$ when the optical density reached a value of 0.8, and washed twice with a 10 mM Tris/HCl Buffer (pH = 7). Cells were then re-suspended in lysis buffer (4% SDS, 0.1 M Tris/HCl, pH 8.0) with a proportion 7.5 $OD_{600}$/100 µl buffer. The cells were disrupted by sonication for 10 min on ice. Unbroken cells were removed by centrifugation at $16,000 \times g$ for 15 min. The protein concentration of the supernatant was determined by bicinchoninc acid (BCA) method (Pierce) and stored at −80 °C.

**Preparation of mouse tissue samples**. Mouse tissues (brain, heart, kidney, liver and lung) used in this study were dissected from mouse strain C57BL/6, males, aged 3 months. The mice were anesthetized with avertin and killed. Tissues were taken out after perfusion with 0.9% NaCl. The procedures were in compliance with ethical regulations and were approved by the ethics committee of Fudan University. All tissues were frozen in liquid nitrogen and stored in −80 °C. We homogenized 50 mg pieces of frozen tissues in 0.4 ml of 4% SDS, 0.1 M Tris/HCl, pH 8.0 using a high-throughput tissue grinding machine (ONEBIO, China) at 65 Hz for 60 s. The crude extract was clarified by centrifugation at $16,000 \times g$ at 30 °C for 40 min. Protein concentration was determined by BCA method.

**Protein digestion**. Proteins were reduced in 10 mM dithiothreitol at 37 °C for 60 min, and then alkylated in dark by 20 mM iodoacetamide at room temperature for 30 min. After carbamidomethylation, the mixture of standard glycoproteins were digested by trypsin. For the proteins from yeast and mouse tissues, six volumes of acetone were added to precipitate the proteins at −20 °C for at least 3 h. The precipitates were dissolved in a denaturing buffer (8 M urea in 50 mM $NH_4HCO_3$) following a ten-fold dilution with 50 mM $NH_4HCO_3$. Trypsin (Promega) was added to a final enzyme-to-substrate ratio of 1:50 and incubated overnight at 37 °C. The reactions were terminated by adding 0.5% trifluoroacetic acid. Finally, all digested samples were centrifuged at $14,000 \times g$ for 10 min and the supernatants were desalted using C18 column (Waters). The desalted samples were then dried by vacuum centrifugation and stored at −20 °C for further use.

**Enrichment of glycopeptides**. Glycopeptides were enriched using zwitterioic hydrophilic interaction liquid chromatography (ZIC-HILIC) described previously with minor modification[7]. Briefly, peptides were loaded onto an in-house ZIC-HILIC micro-column containing 30 mg of ZIC-HILIC particles (Merck Millipore) packed onto a C8 disk. The flow through was collected and passed back through the column for four additional times. The column was washed with 800 µl of 80% acetonitrile and 1% trifluoroacetic acid. Enriched glycopeptides were eluted with 200 µl 0.1% trifluoroacetic acid followed by 20 µl of 25 mM $NH_4HCO_3$ and finally 20 µl of 50% acetonitrile and dried by vacuum centrifugation.

**Liquid chromatography**. The standard glycoprotein mixture, yeast and mouse tissues were analyzed by nanospray LC-MS/MS on an Orbitrap Fusion Tribrid (Thermo Scientific) coupled to an EASY-nano-LC system (Thermo Scientific) without the trap column. For one LC-MS/MS run, 10 µg glycoproteins/100 µg proteins from yeast or mouse tissues were used as starting material (before HILIC enrichment). Samples were loaded onto a C18 spray tip 15 cm × 75 µm i.d. column (standard glycoprotein mixture)/50 cm × 75 µm i.d. column (yeast and mouse tissues) and were separated at a flow rate of 300 nL/min. Solvent A was 0.1% formic acid in water. Solvent B was acetonitrile with 0.1% formic acid. The gradient was 1 h in total for glycoprotein mixture: 5–40% solvent B in 50 min, followed by an increase to 90% B in 3 min and held for 5% B for the last 5 min. The gradient was 6 h in total for complex samples: 5–40% in 345 min, followed by an increase to 90% B in 3 min, held for another 7 min and held for 2% B for the last 5 min.

**Fig. 6** Results of a large-scale intact N-glycopeptide analysis of mouse tissues. **a** Precursor mass deviation of 79,930 GPSMs. **b** Retention times of 61 different glycopeptides with the same peptide backbone 'NLSYEAAPDHK'. The x-axis represents the window of retention time, the y-axis represents the log10 (intensity), and each color represents a different glycopeptide. **c** Example of a chimera spectrum from multiple glycopeptides. pGlyco 2.0 identified two different glycopeptides in one spectrum, illustrated in the form of a mirrored spectral annotation. The top and bottom spectra in each figure are the same spectrum with different glycopeptide identifications. The design of the annotation in each spectrum is the same as that in Fig. 1. **d** The MS1 data corresponds to the chimera spectrum

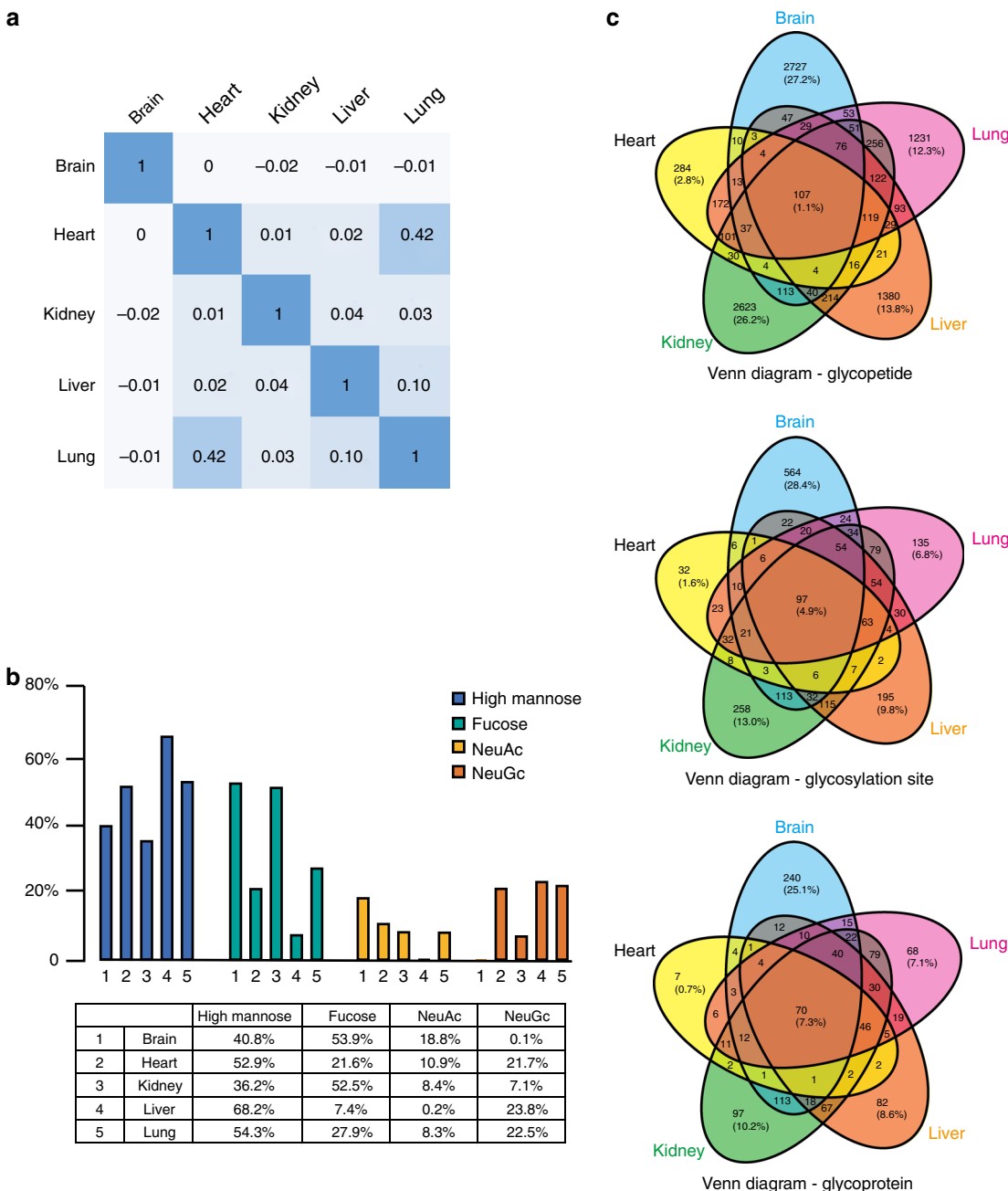

**Fig. 7** Analysis of the glycosylation profiles in different mouse tissues. **a** Correlation analysis of the glycopeptides in different tissues. **b** Distribution of glycopeptides containing high levels of mannose, fucose, NeuAc and NeuGc in different tissues. **c** Venn diagram of the unique glycopeptides, glycosylation sites and glycoproteins in different tissues

**Mass spectrometry analysis for SCE-HCD-MS/MS**. The parameters for glyco-peptide analysis was: (1) MS: scan range ($m/z$) = 800–2000; resolution = 120,000; AGC target = 200,000; maximum injection time = 100 ms; included charge state = 2–6; dynamic exclusion after n times, $n$ = 1; dynamic exclusion duration = 15 s; each selected precursor was subject to one HCD-MS/MS; (2) HCD-MS/MS: iso-lation window = 2; detector type = Orbitrap; resolution = 15,000; AGC target = 500,000; maximum injection time = 250 ms; collision energy = 30%; stepped col-lision mode on, energy difference of ± 10% (10% as absolute value in the Orbitrap Fusion).

**MS data extraction and parameters for database searching**. Raw MS/MS data was converted to 'mgf' format by revised version of pParse[36]. Parameters for database search of intact glycopeptide are as follows: mass tolerance for precursors and fragment ions were set as ± 5 p.p.m. and ± 20 p.p.m., respectively. The protein databases were from Swiss-Prot, version 15.03. The enzyme was full-trypsin.

Maximal missed cleavage was 3. Fixed modification was carbamidomethylation on all Cys residues (C +57.022 Da). Variable modifications contained oxidation on Met (M +15.995 Da), acetylation on protein N-term (+42.011 Da). The N-glycosylation sequon (N-X-S/T, X ≠ P) was modified by changing 'N' to 'J' (the two shared the same mass). The glycan database was extracted from GlycomeDB (www.glycome-db.org), total entries of N-glycan were 7884 by considering NeuGc. The protein databases used were different for each sample:

1) The standard glycoprotein mixtures. Protein database with species of *Homo sapiens* (20,215 entries) was used.
2) Yeast glycoproteome analysis. Protein databases with species of *S. pombe* (4,974 entries) and *Mus musculus* (16,711 entries) were used.
3) Mouse glycoproteome analysis.

Protein database with species of *M. musculus* (16,711 entries) was used.

Different search engines (In our study, pGlyco 2.0 and Byonic) used exactly the same parameter in yeast and mouse glycoproteome analysis. The version of Byonic

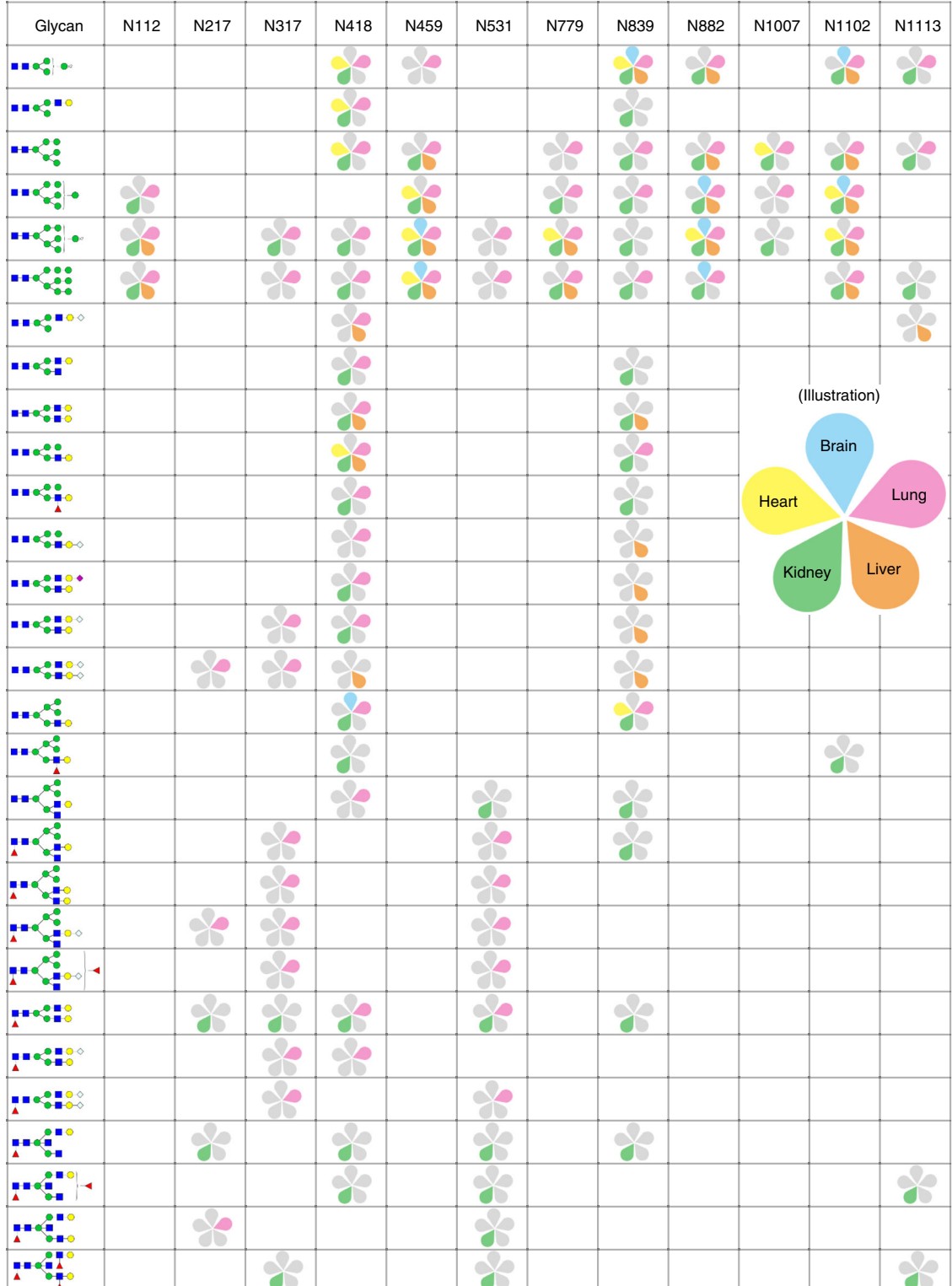

**Fig. 8** Analysis of the glycosylation profile of protein Q3V3R4 (integrin alpha-1) in five mouse tissues. The glycosylation sites are listed in the first row, and the glycans are listed in the first column. The identified site-specific glycans are shown as a petal-shaped mini figure inside the block to demonstrate the tissue specificity

was 2.7.4 that released in Nov. 2015. Default quality control methods for intact glycopeptide identification were 1% GPSM FDR for pGlyco 2.0 and 1% default FDR for Byonic.

**Identification of glycopeptide by pGlyco 2.0.** pGlyco 2.0 was an integrated search engine specifically designed for the interpretation of glycopeptide SCE-

HCD-MS/MS spectra. The procedures of glycopeptide identification in pGlyco 2.0 includes coarse-scoring, fine-scoring and GPSM FDR analysis of glycopeptide.

The first step is coarse-scoring, which was an open search mode for the analysis of the glycan candidates[31]. Given a spectrum, in the coarse-scoring step, for each glycan in the glycome database, the associated peptide backbone mass was calculated as the precursor mass of this spectrum minus the glycan mass, and then the associated masses of all Y ions (glycan fragment ions with peptide backbone

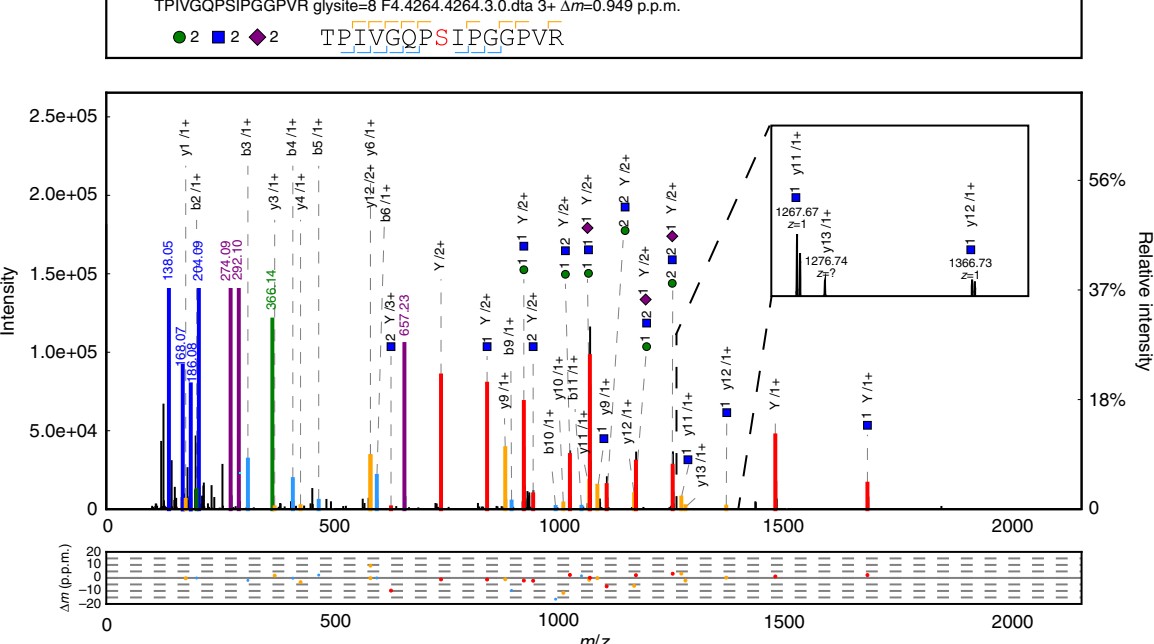

**Fig. 9** Example of an O-glycopeptide SCE-HCD-MS/MS spectrum. A high-quality O-glycopeptide spectrum with abundant glycan and peptide fragment ions derived from stepped-energy HCD-MS/MS is shown. The design of the annotation in each spectrum is the same as that in Fig. 1. This O-glycopeptide has two potential glycosylation sites: T1 and S8. The three diagnostic ions with a HexNAc attached show that S8 is the glycosylation site: y9 + HexNAc ($m/z = 1082.59$, 1 + ), y11 + HexNAc ($m/z = 1267.67$, 1 + ), and y12 + HexNAc ($m/z = 1366.73$, 1+). A zoom-in figure of the latter two fragments is shown in the upper right corner of the spectrum annotation

attached) could be deduced. Each glycan was scored by the number of matched Y ions, and any glycan with less than 2 trimannosyl core ions matched will be filtered out. And the top-k (K = 100 by default) candidate glycans were kept for the fine-scoring step.

The second step is fine-scoring, which was a scoring scheme for a GSM. For each valid glycan candidate after coarse scoring, the corresponding candidate peptides were searched by pFind based only on the peptide backbone mass[37, 38]. pFind has been compiled into a static link library that could be called automatically by pGlyco 2.0 without installing pFind Studio. After peptide search, the candidate glycopeptide candidates were generated by combining glycan candidates and peptide candidates, and fine-scoring was then performed for the GSM to obtain scores for the glycan, peptide and glycopeptide.

The scoring scheme of glycan was based on our previous work[31], which considered the matched peaks, their matching mass errors and the number of matched trimannosyl core ions:

$$\text{Score}_G = \sum_i \log(\text{inten}_i)\left(1 - \left|\frac{\text{merr}_i}{\text{tol}_i}\right|^4\right)(\text{ratio}_\text{ion})^\alpha(\text{ratio}_\text{core})^\beta. \quad (1)$$

In pGlyco 2.0, we have also developed a similar scheme for the scoring of peptide backbone:

$$\text{Score}_P = \sum_i \log(\text{inten}_i)\left(1 - \left|\frac{\text{merr}_i}{\text{tol}_i}\right|^4\right)(\text{ratio}_\text{ion})^\gamma. \quad (2)$$

The meanings of terms are: $\text{ratio}_\text{core}$ = #matched trimannosyl core ions/# theoretical trimannosyl core ions; $\text{ratio}_\text{ion}$ = # matched ions/# theoretical ions; $\text{merr}_i$ is the matching mass error of the $i$-th matched peak; $\text{tol}_i$ is the mass tolerance of the $i$-th matched peak.

The total score of the glycopeptide was the weighted sum of these two scores:

$$\text{Score}_{GP} = w \times \text{Score}_G + (1 - w) \times \text{Score}_P. \quad (3)$$

The four parameters, $\alpha$ and $\beta$ of $\text{Score}_G$, $\gamma$ of $\text{Score}_P$ and $w$ of $\text{Score}_{GP}$ were fine-tuned as $\alpha = 0.56$, $\beta = 0.42$, $\gamma = 0.94$ and $w = 0.35$ by Ranking SVM based on the SCE-HCD-MS/MS spectra. The fine-tuning process of $\text{Score}_G$, was described as an example: for a well-designed fine-scoring scheme, a correct match of a spectrum would always 'beat' other incorrect matches and be ranked as top-1. The aim of fine-tuning the parameters of the scoring scheme was to rank as many correct

matches onto top-1 as possible. The learning-to-rank model was very suitable for this scenario. For $\text{Score}_G$, it was not easy to fine-tune the parameters because the score was an exponential form of the parameters. Taking the logarithm of $\text{Score}_G$ would get a linear form of the parameters, which became:

$$\log(\text{Score}_G) = \log\left(\sum_i \log(\text{inten}_i)\left(1 - \left|\frac{\text{merr}_i}{\text{tol}_i}\right|^4\right)\right) + \alpha \times \log(\text{ratio}_\text{ion})$$
$$+ \beta \times \log(\text{ratio}_\text{core}). \quad (4)$$

This linear form could be easily modeled by Ranking SVM, which is a very popular learning-to-rank algorithm for machine learning. With manual inspection, we could get the correct and incorrect GSMs, and then the Ranking SVM model could be trained on these benchmark GPSMs.

After the coarse-scoring and fine-scoring, pGlyco 2.0 performed GSPM FDR analysis. To our knowledge, there is no widely accepted protocol for FDR analysis of glycopeptide identification in glycoproteomics yet. We carefully studied the false glycopeptide identification and derived a new mathematical model for the FDR analysis of the intact glycopeptide identification. For a GSM, an incorrect identification of either the glycan or the peptide was called a false identification, so the FDR of glycopeptide could be written as:

$$\widehat{\text{FDR}}(x) = p(G = \text{false} \cup P = \text{false}|X \geq x). \quad (5)$$

Here, $G$ = false and $P$ = false refer to the false identification of the glycan and the peptide respectively, and $x$ is the given score threshold. Since $p(G \cup P) = p(G) + p(P) - p(G \cap P)$, Eq. (5) could be rewritten as

$$\widehat{\text{FDR}}(x) = p(G = \text{false}|X \geq x) + p(P = \text{false}|X \geq x)$$
$$- p(G = \text{false} \cap P = \text{false}|X \geq x),$$
$$\widehat{\text{FDR}}(x) = \widehat{\text{FDR}}_G(x) + \widehat{\text{FDR}}_P(x) - \widehat{\text{FDR}}_{G \cap P}(x). \quad (6)$$

For $\widehat{\text{FDR}}_G(x)$, it could be estimated by using our previously reported glycan decoy method coupled with a finite mixture model algorithm[3], and the $\widehat{\text{FDR}}_P(x)$ of peptide could be estimated as #pep_decoy_glycan_target/#both_target, $\widehat{\text{FDR}}_{G \cap P}(x)$ could be estimated by #pep_decoy_glycan_decoy and #both_target.

**FDR analysis of the yeast glycoproteomic dataset**. Estimation of isotope-based FDR: the potential of validating regular peptide identification by metabolic labeling has been introduced by these work[39–41], Zhong et al. emphasized comparing paired labeled and unlabeled fragments in MS/MS spectra (but the pairs are not available most of the time due to stochastic data-dependent acquisition in shotgun proteomics)[39], Snijders et al.[40] and Nelson et al.[41] emphasized comparing paired labeled and unlabeled precursors in the full MS scans or MS1 spectra (the pairs are always available), and made further discussion on how to estimate and reduce false positive identifications[40, 41]. In our pipeline, given a GPSM, a quantification software tool pQuant was employed to find the signal pair of unlabeled and $^{15}$N/$^{13}$C labeled glycopeptide precursors in full MS scans[42]. If no pair was found, or a ratio far away from the targeted 1:1 was obtained, pQuant would output a NaN (Not a Number) ratio for this GPSM, and with high probability, this GPSM was false positive. To estimate FDR, we should estimate the number of all false-positive GPSMs, but not every false-positive GPSM would have an associated NaN ratio. To estimate how sensitive this NaN test can discover false positive GPSMs, we check all decoy GPSMs, which should all be false positive, and see how many of them have associated NaN ratios. Then the isotope-based FDR of a given group of target GPSMs could be estimated by (# NaN in all target GPSMs/# all target GPSMs)/(# NaN in all decoy GPSMs/# all decoy GPSMs). The same procedure was applied to each raw MS data file and each search engine independently. For example, the isotope-based FDR of pGlyco 2.0 in Fig. 2c was calculated as

Replicate 1: $(3/705)/(735/1484) = 0.86\%$
Replicate 2: $(5/713)/(714/1417) = 1.39\%$
Replicate 3: $(2/743)/(681/1380) = 0.55\%$
$(0.86\% + 1.39\% + 0.55\%)/3 = 0.97\%$

Therefore, the average isotope-based FDR of pGlyco 2.0 in three replicate runs was 0.97%.

Estimation of entrapment-based FDR: in the database search process, the proteome and glycome databases of both yeast and mouse were used. The databases of mouse were used as entrapment databases. Any GPSM with a mouse-only peptide or a mouse-only glycan was considered as false positive. For example, if out of 1000 GPSMs in the yeast glycoproteome analysis, 10 GPSMs have mouse-only peptides and another 10 GPSMs have mouse-only glycans, then the entrapment-based FDR in this case was calculated as: $(10 + 10)/1000 = 2.0\%$.

**Analysis of mouse glycoproteomic data set**. Data analysis on retention times: MS1 data was converted from RAW file to '.ms1' format by pParse[36]. The retention times of identified glycopeptides were extracted from MS1 data using pGlyco 2.0 and in-house scripts. A precursor tolerance of ± 5 p.p.m. was used for glycopeptide retention time extraction.

Correlation analysis: site-specific glycosylation comparison among different mouse tissues was performed by calculating Pearson correlation coefficient between the spectrum counts of glycopeptide identification in tissues: each GSM was first converted to site-specific glycosylation information in the format of ('protein-, glycosylation site-, glycan'). After that, the spectrum counts of site-specific glycosylation were calculated and were used as the input data of Pearson correlation coefficient calculation.

**Comparison between pGlyco 2.0 and NGAG/GPQuest**. For the comparison with GPQuest, the glycopeptide identification results of GPQuest were extracted from the Supplementary File from ref. [34]. without any modification. We performed pGlyco 2.0 database search for intact glycopeptide analysis on the raw data in ref. [34]. The search parameters were the same as described in the previous section 'MS data extraction and parameters for database searching' except the proteome database. Two databases were used: (1) de-glycopeptides reported in refs [34].) the complete human proteome database. It was worth mentioning that since SCE was not used in this data set, no adequate glycan fragmentation information was available in MS/MS spectra and therefore we could not perform effective glycan FDR analysis in glycopeptide identification. Here pGlyco 2.0 reported glycopeptide identification using 1% peptide-spectrum matching FDR. All other glycopeptide identifications by pGlyco 2.0 in this paper, which were all based on SCE-HCD-MS/MS, used the more stringent 1% GPSM FDR.

**Data availability**. pGlyco 2.0 program and the source code, as well as a manual for the program:

http://pfind.ict.ac.cn/download/pGlyco/pGlyco2-stable.zip

The RAW MS data of yeast and mouse glycoproteome analysis, as well as the original search results of pGlyco 2.0 have been uploaded to the PRIDE partner repository[43] with access codes:

Yeast: PXD005565
Mouse brain: PXD005411
Mouse heart: PXD005413
Mouse Kidney: PXD005412
Mouse Liver: PXD005553
Mouse Lung: PXD005555

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

## Acknowledgements

The study was supported by grants from the National Key Research and Development Program (2016YFA0501300 to S.-M.H.), National Natural Science Foundation of China (21227805 to P.-Y.Y. and R.-X.S., 31600665 to W.-Q.C., 31570825 to P.-Y.Y., 31300680 to Y.Z. and 31500667 to C.P.), China Postdoctoral Science Foundation (2015M570324 to W.-Q.C.), CAS Interdisciplinary Innovation Team (S.-M.H.), CAS Strategic Pioneer Project (C.C.L.W), CAS Outstanding Technology Talent Program (C.C.L.W.), CAS Key Technology Talent Program (C.P.), National Basic Research Program of China (2012CB910602 to C.L. and 2, 013CB911203 to R.-X.S.), Hi-Tech Research and Development Program of China (2014AA020902 to H.-L.S., P.-Y.Y. and S.-M.H., 2014AA020901 to H.C., 2015AA020104 and 2015AA020108 to Y.Z.), Youth Innovation Promotion Association CAS (No. 2014091 to H.C.) and the International S&T Cooperation Program of China [grant number 2014DFB30010.

## Author contributions

M.-Q.L. generated the experimental design, performed the data analysis and prepared the manuscript. W.-F.Z. performed the data analysis, prepared the manuscript and developed pGlyco 2.0; P.F and W.-Q.C performed wet-laboratory experiments and contributed to the manuscript preparation. C.L. performed the data analysis and contributed to the manuscript preparation. J.-Q.W., X.-J.Z. and Y.Z. contributed to the data analysis and developed pGlyco 2.0; G.-Q.Y. and C.P. performed MS/MS analysis; H.C. and R.-X.S. contributed to pGlyco 2.0 development. Y.C., M.-Q.D., B.-Y.J., J.-M.H. and H.-L.S. contributed to the wet-laboratories experiments. C.C.L.W. contributed to the data collection and interpretation and revised the manuscript; S.-M.H. directed the software development and revised the manuscript. P.-Y.Y. directed the project and revised the manuscript.

## Additional information

**Competing interests:** The authors declare no competing financial interests.

