## [Peer Review File · Nature Communications]

PEER REVIEW FILE

Reviewers' Comments:

Reviewer #1 (Remarks to the Author):

Liu et al provide a manuscript of their new pGlyco2.0 that appears to be a significant step forward in N-glycopeptide identification. Given the major challenge that glycoproteomics provides to proteomic/glycomic investigators, this work is of significant moment. The authors set up a very detailed and well-controlled work flow to demonstrate the power of their approach (Stepped HCD with pGlyco2.0 assignment) and compare it to Byonic (probably the most widely used software program being used for glycan site-mapping on peptides). The FDR calculations used by Byonic have been something that many in the field have been concerned about and the authors clearly demonstrate not only that Byonic is under-representing FDR but that their new program is accurately predicting FDR. It is also laudable that the authors are providing pGlyco2.0 as freeware (many similar programs are commercialized). Finally, the authors apply their approach and software to generate the most comprehensive list of N-linked glycopeptides in mouse tissues to this reviewer's knowledge. Enthusiasm for this manuscript is high with a few minor considerations that should be addressed below and one major suggested additional experiment (3) as further validation of their approach:

1. A careful editing of the manuscript by a native English speaker is necessary and invaluable to convey their exciting findings.
2. The authors should focus this manuscript completely on the generation of high quality spectra and data analysis of N-linked glycopeptides. This should be in the abstract and perhaps even the title. They have not clearly overcome the major hurdle that is O-linked glycopeptides. This is not a criticism but simply a fact and the authors should be clear not to overstate their achievement. The achievement in methodology and software analysis is impressive but has only been clearly demonstrated for N-linked glycopeptides.
3. The NeuGc finding in liver is very surprising (100 times more Gc than Ac structures!!!)--this brings up some concern for the validity of the program in calling Gc. I believe the reviewers should do N-linked glycomics in both brain and liver to validate these results. If the authors are able to see the same enrichments in fucosylated and sialylated structures, this would add considerable validity to their method. It is also quite possible (perhaps Anne Dell's group) that such glycomic analyses have already been performed in the literature and if so should be cited and could avoid the need for detailed glycomics analyses (the reviewer was unable to find such

information in a detailed fashion). Otherwise, I believe glycomics needs to be performed in order to validate these findings.

Reviewer #2 (Remarks to the Author):

The work covers an important and highly relevant aspect of proteomic, namely the post translational modification of glycosylation. There is currently one commercial software (Byonic) that is commonly used for glycoprotein analysis. The software, while more effective than other more commonly used software centered more on protein identification, still has problems when analyzing glycosylation. The pGlyco software addresses these limitations. The authors compare their methods to Byonic and show that their method is generally more effective. Indeed, the results are impressive in the scope of glycosylation and the number of glycopeptides and proteins and show that perhaps we are moving forward with more comprehensive glycoproteomics method.

There are issues that need to be addressed in the manuscript:

1. The grammar is good, but the manuscript needs generally better editing.
2. Proper phrasing would also be useful. For example, in the introduction with the paragraph starting with "To our knowledge...", the paragraph addresses some very important points as to why glycoprotein analysis is different but is difficult to understand. Restating this section would help their argument.
3. This version 2.0 is improved from the first version as stressed in the text. It was then tested on standard protein. This is a good approach, but it would be good to show proteins with known glycosylation sites and determine the sites and number of glycoforms determined. These results are not shown.
4. While the performance of the improved software is good, the manuscript seems to be a mishmash of results using different methods. The presentation here can be better and perhaps made more focused rather than an application of the method on a number of unrelated efforts.
5. An example of number of proteins with the glycoforms delineated for specific proteins would be useful and would go a long way towards showing that there are actual structures and not just an increasing number of peptides.

**(POINT-BY-POINT REPLY)**

Current Figure		Original Figure	Content
a	1a	Glycopeptide spectra with optimized stepped energy
	b	1b	Glycopeptide spectra with default energy
1c	Workflow of pGlyco 2.0
New	Site-specific glycan demonstration of standard glycoproteins
a	2a	Validation workflow using isotope labeling
	b	2b	Principle of two FDR evaluation methods
a	S7-2a	Analysis of isotope pairs of glycopeptide reported by pGlyco 2.0
	b	S7-2b	Analysis of isotope pairs of glycopeptides reported by Byonic
	c	2c	FDR analysis of glycopeptide-spectrum matching from pGlyco 2.0 and Byonic
	d	S7-3	FDR analysis of Byonic under different score thresholds
a	3a	Precursor mass accuracy of glycopeptides in the mouse data
	b	3b	Retention time of glycopeptides with different numbers of sialic acids
	c	S8-3	Example chimera glycopeptide spectrum
	d	S8-4	MS1 profile corresponding to the chimera spectrum
a	3c	Correlation of glycosylation profile in mouse tissues
	b	3d	Glycan distribution of glycopeptides in mouse tissues
	c	S13	Venn diagram of glycosylation information in mouse tissues
New	Site-specific glycan demonstration of the protein Integrin alpha-1 in mouse tissues
S16	Example of O-glycopeptide spectrum

1 Figure 1. Position and content of figures in the revised manuscript.

2

Point-by-point reply to reviewer#1

Q: Liu et al provide a manuscript of their new pGlyco2.0 that appears to be a
significant step forward in N-glycopeptide identification. Given the major
challenge that glycoproteomics provides to proteomic/glycomic investigators,
this work is of significant moment. The authors set up a very detailed and
well-controlled work flow to demonstrate the power of their approach (Stepped
HCD with pGlyco2.0 assignment) and compare it to Byonic (probably the most
widely used software program being used for glycan site-mapping on peptides).
The FDR calculations used by Byonic have been something that many in the
field have been concerned about and the authors clearly demonstrate not only
that Byonic is under-representing FDR but that their new program is accurately
predicting FDR. It is also laudable that the authors are providing pGlyco2.0 as
freeware (many similar programs are commercialized). Finally, the authors apply
their approach and software to generate the most comprehensive list of
N-linked glycopeptides in mouse tissues to this reviewers knowledge.
Enthusiasm for this manuscript is high with a few minor considerations that
should be addressed below and one major suggested additional experiment (3)
as further validation of their approach.

22 A: Thank you for the praise of our method. We have addressed these minor
considerations, including the language editing and method introduction. For
the suggested additional experiment, we have performed not only glycomic
analysis but also other complementary techniques to validate our results.

Q: A careful editing of the manuscript by a native english speaker is necessary
and invaluable to convey their exciting findings.

32 A: English language editing has been provided by the Nature Research Editing
Service.

Q: The authors should focus this manuscript completely on the generation of
high quality spectra and data analysis of N-linked glycopeptides. This should be

in the abstract and perhaps even the title. They have not clearly overcome the
major hurdle that is O-linked glycopeptides. This is not a criticism but simply a
fact and the authors should be clear not to overstate their achievement. The
achievement in methodology and software analysis is impressive but has only
been clearly demonstrated for N-linked glycopeptides.

7 A: We agree with this point and have made changes to our manuscript
accordingly, for example:

1) The title is “pGlyco 2.0 enables precision **N-glycoproteomics** with
comprehensive quality control and one-step mass spectrometry for intact
glycopeptide identification” .

2) In the abstract, we introduce our work as “we propose a workflow for the
precise identification of intact **N-glycopeptides** at the proteome scale and in a
high-throughput manner using stepped-energy fragmentation in mass
spectrometry and a dedicated search engine, pGlyco 2.0” .

3) In the discussion, we conclude our work as “we have presented a dedicated
workflow that combines a fine-tuned, easily adopted MS protocol and a
dedicated search engine, pGlyco 2.0, which are ready for use in the precise
**N-glycoproteomic** analysis of complex samples” .

Q: The NeuGc finding in liver is very surprising (100 times more Gc than Ac
structures!!!)--this brings up some concern for the validity of the program in
calling Gc. I believe the authors should do N-linked glycomics in both brain and
liver to validate these results. If the authors are able to see the same
enrichments in fucosylated and sialyated structures, this would add
considerable validity to their method. It is also quite possible (perhaps Anne
Dell's group) that such glycomic analyses have already been performed in the
literature and if so should be cited and could avoid the need for detailed
glycomics analyses (the reviewer was unable to find such information in a
detailed fashion). Otherwise, I believe glycomics needs to be performed in
order to validate these findings.

35 A: We have performed analyses using four different methods to validate our
novel NeuGc finding in mouse liver, which include the following:

1) Glycomics analysis of mouse liver and brain.

2) NeuAc and NeuGc profiling through DMB labeling and UHPLC analysis.

3) Search engine-independent, bioinformatics analysis of RAW data from
glycopeptide analysis.

4) Information from the literature.

We will report our results in detail here.

		High mannose	Fucose	NeuAc	NeuGc
Brain	40.8%	53.9%	18.8%	0.1%
Heart	52.9%	21.6%	10.9%	21.7%
Kidney	36.2%	52.5%	8.4%	7.1%
Liver	68.2%	7.4%	0.2%	23.8%
Lung	54.3%	27.9%	8.3%	22.5%

Figure 2. Distribution of high mannose-, fucose-, NeuAc- and
NeuGc-containing glycopeptides in different tissues. This is the same figure as
Figure 7-b in the manuscript. The reviewer questioned the surprising ratio
between NeuAc and NeuGc in the mouse liver.

**1. Glycomics analysis on mouse liver and brain**

We established an MS-based glycomic analytical pipeline to quantitatively
analyze the N-glycome of mouse liver and brain. The glycomics data, which
contained both MS and MS/MS information, revealed 17 times more Gc than Ac
structures in mouse liver. As validation of our glycomic pipeline, we found trace
amounts of Gc structures in mouse brain. These glycomic data were consistent
with our glycopeptide analysis.

In the glycomic analysis, we used the same LC and MS instrument (an Orbitrap
Fusion Tribrid coupled to an EASY-nano-LC system without the trap column) as
that used in the glycopeptide analysis. The LC column was different, a PGC 15
14 cm x 75 μm i.d. column (New Objective) was used for N-glycan analysis. The
15 gradient lasted 30 min: 1% to 40% solvent B in 20 min, followed by an increase
to 85% B in 1 min and a hold for 9 min. The MS parameters were the same as
those used in the glycopeptide analysis, except that we fine-tuned the MS/MS
collision conditions: we used HCD-MS/MS with a stepped energy of 25% \pm 10%
and CID-MS/MS with an energy of 30% for the tandem-MS analysis of each
precursor. In the glycan-spectrum-matching procedure, the HCD-MS/MS and
CID-MS/MS spectra of each precursor were merged into a single spectrum.

We used an unpublished, automated glycomic search engine Glyconote to
analyze the MS data. This software is a result of our collaboration with another
well-established glycomic lab. Glyconote performed high-throughput open
searching of all MS/MS spectra, automated quality control and spectral
annotation. After that, Glyconote could quantify the abundance of the
identified glycans in MS1 data.

Below are some annotated spectra from our glycomic analysis of the mouse
tissues.

Composition: 8_2_0_0_0 (Hex HexNAc dHex NeuAc NeuGc) RT: 7.781 Precursor: 1721.6007
 Combination: Hex 8 HexNAc 2 H2O_1
 Title: Gly-L-R1.1318.1318.2.0.dta
 Cov. Int : 88% Cov. Seq : 88%

- 1 Figure 3. Annotated MS/MS spectra of a glycan with a composition of Hex X 8 +
- 2 HexNAc X 2. Spectral information is on the top, and matched peaks are shown
- 3 in red with an annotation of "composition, # of monosaccharides, charge, m/z
- 4 and relative intensity" . Unmatched peaks are shown in black.

Composition: 5_4_0_0_1 (Hex HexNAc dHex NeuAc NeuGc) RT: 14.681 Precursor: 1948.6907
 Combination: Hex 5 HexNAc 4 NeuGc_1 H2O_1
 Title: Gly-L-R1.3439.3439.2.0.dta
 Cov. Int : 80% Cov. Seq : 88%

- 5 Figure 4. Annotated MS/MS spectra of a glycan with a composition of Hex X 5,
- 6 HexNAc X 4, and NeuGc X 1.

The following four figures show the MS1 elution profile of the glycans identified
in mouse liver.

Figure 5. MS1 elution profile of glycans, 7.0~8.0 min.

Figure 6. MS1 elution profile of glycans, 8.6~9.7 min.

1 Figure 7. MS1 elution profile of glycans that contain NeuGc, 12.0~22.0 min.

2

3 Figure 8. MS1 elution profile of glycans that contain NeuAc, 12.0~22.0 min.

4

From the quantitative information demonstrated in Figures 7 and 8, it is clear
that the majority of sialic acids in the N-glycome of the mouse liver are NeuGc.
The total intensity of NeuGc-containing glycans was 5.2×10^8 , while that of
NeuAc-containing glycans was 3.0×10^7 . Therefore, there were approximately 17
5 times more NeuGc than NeuAc structures in the glycome data.

We also performed a glycomic analysis of mouse brain. The ratio of NeuAc to
NeuGc agreed with the glycopeptide data reported by pGlyco 2.0. [REDACTED]

19

**2. NeuAc and NeuGc profiling through DMB labeling and UHPLC analysis**

We analyzed the amount of NeuAc and NeuGc in different mouse tissues using
a commercial sialic acid releasing/labeling kit. The sialic acids from
glycoproteins were released and specifically labeled using a LudgerTag™ DMB
(1,2-diamino-4,5-methylenedioxybenzene.2HCl) Kit. Then, the DMB-labeled
sialic acids were identified and relatively quantitated using reversed-phase
chromatography. First, a DMB-labeled sialic reference panel (containing
Neu5Ac, Neu5Gc, Neu5,7Ac2, Neu5,Gc9Ac and Neu5,9Ac2) and NeuAc and
NeuGc standard monosaccharide were analyzed as standards (Supplementary
Figure 12-a to 12-c). Then, human IgG, which only contains Neu5Ac, and bovine
fetuin, which contains predominantly Neu5Ac and a small amount of NeuGc
(2-3%), were analyzed as positive controls (Supplementary Figure 12-d, e).
Ovalbumin, which contains only trace amounts of sialic acids, was analyzed as a
negative control (Supplementary Figure 12-f). The analysis results were highly
consistent with existing knowledge. Finally, DMB-labeled sialic acids from
mouse liver, brain and kidney were analyzed to determine the relative amounts
of Neu5Ac and Neu5Gc in each tissue. Trace amounts of NeuAc were detected
in the liver (Supplementary Figure 12-g), while the opposite distribution of sialic
acids was observed in the brain (Supplementary Figure 12-h). Meanwhile, the
amounts of NeuGc and NeuAc in the kidney were similar (Supplementary
Figure 12-i). The above analysis results on the abundance of NeuAc and NeuGc
in mouse tissues were consistent with our glycopeptide data obtained using
pGlyco 2.0.

To determine whether the kit had completely released the sialic acids in the
sample, we also performed glycopeptide analysis of the sample after release.
pGlyco 2.0 did not identify any glycopeptides with sialic acids in mouse liver,
brain or kidney.

Figure 9. Results of the NeuAc and NeuGc analysis using a sialic acid
 releasing/labeling kit. This is the same figure as Supplementary Figure 12 in the
 manuscript. The abundance of NeuAc and NeuGc in different samples. (a)
 Reference panel. (b) Standard NeuGc. (c) Standard NeuAc. (d) IgG from human.
 (e) Bovine fetuin. (f) Ovalbumin. (g) Mouse liver. (h) Mouse brain. (i) Mouse
 kidney.

**3. Search engine-independent, bioinformatics analysis of RAW data from**
**glycopeptide analysis**

Many monosaccharides can generate diagnostic ions in the MS/MS spectra of
glycans or glycopeptides. NeuAc and NeuGc will generate distinct diagnostic
ions of 292.1 and 307.1, respectively. The abundance of the diagnostic ions
should be an indicator of the amount of the corresponding monosaccharide in
the sample.

We therefore analyzed the intensity of 292.1 and 307.1 in all raw MS/MS spectra
from the mouse data. This process is search engine independent; thus, the
result can serve as a validation of the glycopeptides reported by pGlyco 2.0. The
results are shown in the following figures. The relative abundances of NeuAc
and NeuGc in mouse liver, brain and kidney agree with the result reported by
pGlyco 2.0.

- 1 Figure 10. Intensity of sialic acid diagnostic markers in the raw MS/MS data
- 2 from mouse liver, brain and kidney. The intensity is the original value reported
- 3 by the MS instrument.
- 4

**4. Information from the literature**

We searched the literature and could not find a detailed MS/MS analysis of the
mouse liver glycome. However, some studies focused on a few glycoproteins in
mouse liver. These studies showed that most glycans with sialic acid on these
glycoproteins have NeuGc, which agrees with our findings. The following
presents two examples:

Figure 11. Original Figure 4 in reference "Medzihradzky, K.F., Kaasik, K. &
Chalkley, R.J. Characterizing sialic acid variants at the glycopeptide level.
Analytical chemistry 87, 3064-3071 (2015)", which shows the superimposed
extracted ion chromatograms of seven different glycoforms of the N-linked
glycopeptide "VVLPNHSVVDIGLIK" from haptoglobin in mouse liver. NeuGc
is the major sialic acid (cyan diamonds), and NeuAc is the minor sialic acid
(purple diamonds).

Figure 12. Original Figure 3 in reference "Medzihradzsky, K.F., Kaasik, K. &
Chalkley, R.J. Tissue-Specific Glycosylation at the Glycopeptide Level. *Molecular*
& *Cellular Proteomics* 14, 2103-2110 (2015)", which shows the glycoforms
detected in mouse brain and liver for 9 N-glycosylation sites of prolow-density
lipoprotein receptor-related protein 1. The glycans found in mouse liver have
large amounts of NeuGc; no NeuAc-containing glycan was reported.

Point-by-point reply to reviewer#2

Q: The work covers an important and highly relevant aspect of proteomic,
namely the post translational modification of glycosylation. There is currently
one commercial softwarey(Byonic) that is commonly used for glycoprotein
analysis. The software, while more effective than other more commonly used
software centered more on protein identification, still has problems when
analyzing glycosylation. The pGlyco software addresses these limitations. The
authors compare their methods to Byonic and show that their method is
generally more effective. Indeed, the results are impressive in the scope of
glycosylation and the number of glycopeptides and proteins and show that
perhaps we are moving forward with more comprehensive glycoproteomics
method.

16 A: Thank you for the praise of our method. We have revised the manuscript to
17 better present our work. The revised work includes language editing, a
18 reorganized manuscript format and the addition of new figures as you
suggested.

Q: The grammar is good, but the manuscript needs generally better editing.

25 A: English language editing has been provided by the Nature Research Editing
Service.

Q: Proper phrasing would also be useful. For example, in the introduction with
the paragraph starting with "To our knowledge..." , the paragraph addresses
some very important points as to why glycoprotein analysis is different but is
difficult to understand. Restating this section would help their argument.

34 A: Thank you for your suggestion. We have restated this paragraph.

Q: This version 2.0 is improved from the first version as stressed in the text. It
was then tested on standard protein. This is a good approach, but it would be

*good to show proteins with known glycosylation sites and determine the sites*
*and number of glycoforms determined. These results are not shown.*

4 A: We have presented the results of the standard glycoprotein analysis in the
5 revised manuscript in three parts: information on all glycopeptide-spectrum
matches is shown in the Supplementary Data; glycans found on multiple
glycosylation sites are shown in Figure 3, and the corresponding annotated
spectra are shown in the Supplementary File; and a comparison of our results
with existing analytical data on these proteins are shown in the Supplementary
Table.

*Q: While the performance of the improved software is good, the manuscript*
*seems to be a mishmash of results using different methods. The presentation*
*here can be better and perhaps made more focused rather than an application*
*of the method on a number of unrelated efforts.*

19 A: We have reorganized the manuscript for better presentation. The original
format was for a "Brief Communication" , which does not include subheadings
in the manuscript (our manuscript was transferred from another journal to
*Nature Communications*). The revised manuscript complies with the format
requirements of *Nature Communications* and has subheadings in the results
section. We believe that the revised manuscript presents our work in a clear and
focused manner.

The results section is divided into four parts:

Part 1: method introduction, which includes the sections "development of an
optimal MS/MS fragmentation method and a dedicated search engine" and
"analysis of a standard glycoprotein mixture" .

Part 2: validation of the proposed method, which includes the sections "FDR
validation workflow" and "FDR validation results" .

Part 3: an application of the site-specific glycosylation analysis to five mouse
tissues, which includes the sections "optimization of LC-MS/MS parameters
for large-scale study" , "Results of a large-scale intact N-glycopeptide analysis

of mouse tissues” and “analysis of the glycosylation profile in different mouse
tissues” .

Part 4: method comparison and potential application, which includes the
sections “comparison with existing large-scale glycoproteome research” and
“application to O-glycopeptide analysis” .

Q: An example of number of proteins with the glycoforms delineated for
specific proteins would be useful and would go a long way towards showing
that there are actual structures and not just an increasing number of peptides.

14 A: We specifically designed a way to present site-specific glycosylation in
different mouse tissues. A new figure (Figure 8) is added to the revised
manuscript, and we would like to briefly discuss the figure here:

Figure 13. This is the same figure as Figure 8 in the manuscript. *Analysis of the*
*glycosylation profile of protein Q3V3R4 (integrin alpha-1) in five mouse tissues.*
*The protein names and glycosylation sites are listed in the first row, and the*
*glycans are listed in the first column. The identified site-specific glycans are*
*shown as a petal-shaped mini figure inside the block to demonstrate the tissue*
*specificity.*

In our glycopeptide data from five mouse tissues, site-specific glycosylation
was analyzed at three dimensions: the protein glycosylation site, glycan and
different tissues. We incorporated this three-dimensional information into a
single figure, resulting in this figure, in which all information is clearly depicted.
Each block in the figure represents a site-specific glycan, and the corresponding
distribution in the tissues can be easily observed in the petal-shaped mini
figure inside the block. In addition, the corresponding annotated spectrum of
each site-specific glycan in the figure is shown in the Supplementary File.

The figure shown here was manually generated. We are developing an
automated way to generate this type of figure. It will be very useful to generate
a figure for each glycoprotein identified by pGlyco 2.0 in the format presented
here. Users could easily screen interesting glycoproteins by viewing the
site-specific glycosylation profile of each glycoprotein.

Reviewers' Comments:

Reviewer #1 (Remarks to the Author):

The authors have been extremely responsive to all suggestions, comments and concerns. Thus, enthusiasm for this revised manuscript is high.

Reviewer #2 (Remarks to the Author):

The authors have produced a convincing point-by-point response to the reviewers' comments. The addition of Figure 8, in particular, is helpful as it shows the variation in glycosylation of protein in various tissues. The manuscript would be a great addition to the field and glycoproteomic analysis, specifically.